# Carbon Monoxide Diffusing Capacity (DL_CO_) Correlates with CT Morphology after Chemo-Radio-Immunotherapy for Non-Small Cell Lung Cancer Stage III

**DOI:** 10.3390/diagnostics12051027

**Published:** 2022-04-19

**Authors:** Markus Stana, Brane Grambozov, Christoph Gaisberger, Josef Karner, Elvis Ruznic, Johannes Berchtold, Barbara Zellinger, Raphaela Moosbrugger, Michael Studnicka, Gerd Fastner, Felix Sedlmayer, Franz Zehentmayr

**Affiliations:** 1Department of Radiation Oncology, Paracelsus Medical University, SALK, 5020 Salzburg, Austria; m.stana@salk.at (M.S.); b.grambozov@salk.at (B.G.); c.gaisberger@salk.at (C.G.); j.karner@salk.at (J.K.); e.ruznic@salk.at (E.R.); j.berchtold@salk.at (J.B.); g.fastner@salk.at (G.F.); f.sedlmayer@salk.at (F.S.); 2Institute of Pathology, Paracelsus Medical University, SALK, 5020 Salzburg, Austria; b.zellinger@salk.at; 3Department of Pneumology, Paracelsus Medical University, SALK, 5020 Salzburg, Austria; r.moosbrugger@salk.at (R.M.); m.studnicka@salk.at (M.S.)

**Keywords:** non-small cell lung cancer, dose volume histogram, carbon monoxide diffusing capacity, high dose radiation, radiation induced lung disease

## Abstract

Introduction: Curatively intended chemo-radio-immunotherapy for non-small cell lung cancer (NSCLC) stage III may lead to post-therapeutic pulmonary function (PF) impairment. We hypothesized that the decrease in global PF corresponds to the increase in tissue density in follow-up CTs. Hence, the study aim was to correlate the dynamics in radiographic alterations to carbon monoxide diffusing capacity (DL${}_{\mathrm{CO}}$) and FEV${}_{1}$, which may contribute to a better understanding of radiation-induced lung disease. Methods: Eighty-five patients with NSCLC III were included. All of them received two cycles of platinum-based induction chemotherapy followed by high dose radiation. Thereafter, durvalumab was administered for one year in 63/85 patients (74%). Pulmonary function tests (PFTs) were performed three months and six months after completion of radiotherapy (RT) and compared to baseline. At the same time points, patients underwent diagnostic CT (dCT). These dCTs were matched to the planning CT (pCT) using RayStation^®^ Model Based Segmentation and deformable image registration. Differential volumes defined by specific isodoses were generated to correlate them with the PFTs. Results: In general, significant correlations between PFTs and differential volumes were found in the mid-dose range, especially for the volume of the lungs receiving between 65% and 45% of the dose prescribed (V${}_{65-45\%}$) and DL${}_{\mathrm{CO}}$ (p<0.01). This volume range predicted DL${}_{\mathrm{CO}}$ after RT (*p*-value 0.03) as well. In multivariate analysis, DL${}_{\mathrm{CO}}$ (*p*-value 0.040) and FEV${}_{1}$ (*p*-value 0.014) predicted pneumonitis. Conclusions: The current analysis revealed a strong relation between the dynamics of DL${}_{\mathrm{CO}}$ and CT morphology changes in the mid-dose range, which convincingly indicates the importance of routinely used PFTs in the context of a curative treatment approach.

## 1. Introduction

Several models have been proposed for predicting radiation induced lung disease, e.g., [1,2,3,4]; however, there remain no clinically verified models that quantitatively predict lung function impairment after radiotherapy (RT) [5,6]. The reason for this is that the reported injury rate is highly dependent on the endpoint considered [7]. In this context, predictive clinical factors for the extent of fibrosis following RT can help in terms of patient selection, while dosimetric parameters can guide plan optimization [8].

Pulmonary function tests (PFTs) are generally considered to be good surrogate markers for assessing radiation-induced pulmonary impairment [5] as they can be assessed more objectively than clinical endpoints [5]. Setup inaccuracies and patient compliance may bias PFTs; a reproducibility within a ±10% margin has been reported [9]. Nevertheless PFTs are regarded as a clinically established method for assessing lung function [10].

When comparing the density values of CTs acquired at different time points, it has been shown that an increase in density occurs in areas with a radiation dose of more than 6 Gy [11]. Although CT density changes after thoracic irradiation are histopathologically correlated with lung fibrosis [12,13], it is not completely clear how these changes are related to lung function parameters. A previous study [7] found only a weak correlation between CT density and PFT dynamics before and after radiation treatment, which led the authors to question the usefulness of lung function testing prior to RT in general.

The aim of the present retrospective study was to correlate the dynamics of carbon monoxide diffusing capacity (DL${}_{\mathrm{CO}}$) and FEV${}_{1}$, which are the most frequently used lung function parameters in clinic, with those of CT density. In accordance with Ma et al. [7], we hypothesized that a decline in PFT values would be correlated with an increase in lung tissue density. The focus of the analysis was on differential volumes encompassed by specific isodoses rather than on total lung volume. By clarifying the relation between CT-morphology and lung function dynamics, this study is intended to contribute to better understanding of the development of radiation-induced lung disease.

## 2. Materials and Methods

### 2.1. Patients

Eighty-five patients who underwent thoracic radiation treatment with curative intent between October 2015 and October 2020 were included. All met the following inclusion criteria: (1) the primary tumor was located in the lung and classified as inoperable; (2) the tumor was histologically or cytologically verified and categorized as stage III according to the 8th edition of the TNM classification; (3) 18F-FDG-PET-CT and cranial MRI were mandatory requirements in the diagnostic work-up; (4) performance status had to be 0–1 according to the Eastern Cooperative Oncology Group (ECOG); (5) patients received curatively-intended chemo-radiotherapy with or without immunotherapy after discussion in a multidisciplinary tumor board with pneumologists, medical oncologists, radiologists, thoracic surgeons, pathologists, and radiation oncologists; (6) at each follow-up visit, a contrast-enhanced chest CT and PFTs including whole-body plethysmography, blood gas analysis, and DL${}_{\mathrm{CO}}$ were performed. The study was approved by the ethics committee of the Federal State of Salzburg (No. 415-E/1915/12-2015).

### 2.2. Treatments: Chemotherapy, Irradiation, Immune Checkpoint Inhibition

First, all patients received two cycles of either cisplatin (75 mg/m${}^{2}$/d) in combination with pemetrexed (500 mg/m${}^{2}$/d) or gemcitabine (1000 mg/m${}^{2}$/d) as induction chemotherapy before irradiation. In case of renal dysfunction, carboplatin (AUC 5 d1) (absolute maximum dose 1100 mg) replaced cisplatin. Second, accelerated high-dose thoracic irradiation was administered with total doses of 66 Gy in 3 Gy fractions or 73.8 Gy in twice-daily fractions of 1.8 Gy as described in a previous publication [14]. Because these radiation treatment schedules differ from each other, the physical doses were converted to EQD${}_{2}$ with an α/β = 10 assumed for tumors, *D* as the total physical dose, and *d* as the dose per fraction.
(1)$${\mathrm{EQD}}_{2}=D\;\frac{d+\alpha /\beta }{2+\alpha /\beta }$$

As for dose constraints, the following limits were applied [2,15,16]: mean lung dose (MLD) < 20 Gy, less than 40% of both lungs was allowed 20 Gy or more (V20_total lung_ < 40%), mean esophageal dose (MED) < 34 Gy, maximum dose to the spinal cord 45 Gy, and less than 10% of the heart was allowed 25 Gy or more (V25${}_{\mathrm{heart}}$ < 10%). Third, as of September 2017 patients received durvalumab (Imfinzi^®^) 10 mg/kg maintenance therapy for one year after the end of RT [17]. Durvalumab is a monoclonal antibody that blocks programmed death ligand 1 (PD-L1), thereby enabling T cells to better recognize the tumor. It is used in the adjuvant treatment of stage III NSCLC after chemo-radiotherapy.

### 2.3. Pulmonary Function Parameters

Thoracic radiation can cause lung tissue changes which in severe cases can lead to clinically relevant pulmonary function loss as a result of fibrosis. The parameters most frequently used in clinic for assessing treatment-related lung tissue changes are FEV${}_{1}$ and DL${}_{\mathrm{CO}}$. In this context, FEV${}_{1}$ is used as a surrogate marker for the narrowing of large or medium-sized bronchi as well as the bronchiolar airways, while DL${}_{\mathrm{CO}}$ represents changes in the alveolar compartment. PFTs were performed before irradiation, and again three months and six months thereafter.

### 2.4. CT Morphology Changes

Changes in lung density were evaluated by means of three diagnostic CT (dCT) datasets, performed three month before RT ($t_{\mathrm{pre}}$) as well as three ($t_{3m}$) and six months ($t_{6m}$) after treatment completion. While it should be noted that CT density and fibrosis are two different entities, in the context of thoracic radiation treatment it is plausible to assume that the anatomical substrate of CT density increase is lung tissue fibrosis. The lung was divided into regions according to the total dose received (Figure 1). In order to determine the different doseregions in the lungs on each of the three CTs, scripting in RayStation^®^ was used.

The body outline was contoured based on the Hounsfield unit (HU) histogram (i.e., Gray Level Threshold in RayStation^®^) for each dCT dataset. The planning CT (pCT) was matched to each of the three diagnostic datasets in the automated manner described below. First, rigid registration focused on bony structures and subsequently included all tissues with equal weight. This procedure was followed by a deformable registration in order to improve the accuracy of image alignment. Second, both lungs were contoured using RayStation^®^ Model-Based Segmentation with 100 iterations for each dCT (RaySearch^®^ deformable registration in RayStation^®^ available online: www.raysearchlabs.com accessed on 30 March 2021).

Hybrid deformable image registration was performed by means of the automatically constrained deformation algorithm (ANACONDA) [18]. The deformation strategy was based on internal lung, correlation coefficient was chosen as a similarity measure, and both lungs were used as a focus ROI and control. To improve comparability, a reference lung ROI including both lungs was created on the pCT and copied to the three dCTs via deformable registration. Third, dose volumes in the lungs were created on pCT with reference to the dose that was administered to the planning target volume (PTV). Total volumes were generated in steps of 10% decrements ranging from V${}_{105\%}$ (=the volume receiving at least 105% of the prescribed dose) to V${}_{5\%}$ (=the volume receiving at least 5% of the prescribed dose). Moreover, differential volumes were generated by subtracting these absolute volumes, for example, V${}_{95-15\%}$, which is the volume that received between 95% and 15% of the prescribed dose (Figure 1c). These absolute and differential volumes were copied to dCTs. Finally, the average density value in HU for each dose volume was read out for further processing.

As lung density depends on scanning protocol parameters such as lung volume, patient positioning, and the use of radiocontrast agents, each dataset was normalized to the whole lung volume. A relative density number ($n_{\rho }$) was introduced by dividing the average CT HU number of the respective total or differential volume (${\bar{HU}}_{\mathrm{Reference}}\;\mathrm{Volume}$) by the average CT HU number of the total lung (${\bar{HU}}_{\mathrm{Total}}\;\mathrm{Lung}\;\mathrm{Volume}$) as provided by Equation (Equation 2):(2)$$n_{\rho }=\frac{{\bar{HU}}_{\mathrm{Reference}}\;\mathrm{Volume}}{{\bar{HU}}_{\mathrm{Total}}\;\mathrm{Lung}\;\mathrm{Volume}}.$$

Of note, the average HU ($\bar{HU}$) of the total lung volume is typically smaller than that of the volume defined by a specific isodose, as the former is more transparent than the latter. Therefore, while a low negative HU number represents low density, $n_{\rho }$ has a positive value (typically smaller than 1) for transparent tissue and decreases to negative values with increasing lung tissue density.

### 2.5. Statistics

Based on the published literature [7,10,11,19,20,21] a positive correlation between lung density changes and PFT dynamics can be assumed, which means that the zero hypothesis for our analysis is a negative or non-correlation. We interpreted the PFT data as a continuous variable. In order to correlate the PFTs (pPFT(t)) for the three time points mentioned above, namely, $t_{pre}$ = three months before, $t_{3m}$ = three and $t_{6m}$ = six months after the end of RT with the relative density number representing the normalized HUs for the same points in time ($n_{\rho }\left(t\right)$), the Pearson correlation coefficient was calculated according to Equation (Equation 3). The average over all three time points yields the average PFT (${\bar{p}}_{\mathrm{PFT}}$) and average relative density number (${\bar{n}}_{\rho }$).
(3)$$r=\frac{\sum_{t=t_{\mathrm{pre}},t_{3m},t_{6m}}\left(n_{\rho }\left(t\right)-{\bar{n}}_{\rho }\right)\left(p_{\mathrm{PFT}}\left(t\right)-{\bar{p}}_{\mathrm{PFT}}\right)}{\sqrt{\sum_{t=t_{\mathrm{pre}},t_{3m},t_{6m}}{\left(n_{\rho }\left(t\right)-{\bar{n}}_{\rho }\right)}^{2}\sum_{t=t_{\mathrm{pre}},t_{3m},t_{6m}}{\left(p_{\mathrm{PFT}}\left(t\right)-{\bar{p}}_{\mathrm{PFT}}\right)}^{2}}}$$

As the Pearson correlation coefficient is limited to the range between −1 and 1, a normal distribution of samples cannot be assumed. Therefore, Fisher z-transformation was performed to allow a one-sided, right-handed *t*-test to be applied in order to test for significance; *p*-values < 0.01 and <0.05 were regarded as highly and moderately significant, respectively. Bonferroni correction was used to correct for multiple testing. Clinical endpoints such as local control (LC), progression-free survival (PFS), and overall survival (OS) were calculated with the Kaplan–Meier-method. Multivariate analysis (MVA) was performed using forward stepwise Cox Regression.

## 3. Results

### 3.1. Patients

Fifty-nine (69.5%) patients were male and 26/85 (30.5%) were female. The median age was 66 years (range: 46–81). The ECOG performance score was 0–1 in 83/85 (98%) patients with a median Charlson Comorbidity Index (CCI) of 5 (range 2–9). Thirty-nine of the 85 patients (46%) had at least COPD grade 1, including 5% with grade 4. With a median FEV${}_{1}$ of 2.2 L (range: 0.8–3.8) and a median corrected DL${}_{\mathrm{CO}}$ of 5.3 mmol/min·kPa (range: 1.4–11.0), all patients had sufficient lung function to undergo curative intent thoracic radiation. For DL${}_{\mathrm{CO}}$ there were 231 measurements available, while for FEV${}_{1}$ 250 were available. Patient data are summarized in Table 1.

### 3.2. Radiation Treatment and Systemic Therapy

In 41/85 (48%) of the patients the tumor was located centrally, with a median size of 19 mL (range 0.3–308). All patients received high-dose irradiation, with a median EQD${}_{2}$ of 72.3 Gy (58.3–88.2) to the tumor. Sixty-four patients received dose-differentiated accelerated radiotherapy (DART) as described previously [14], with total radiation doses between 73.8 Gy and 90 Gy depending on tumor size. As of January 2020 patients had received 66 Gy in 3 Gy fractions, which is biologically in the same range as 73.8 Gy in 1.8 Gy fractions. This dosage was administered in intensity-modulated radiotherapy (IMRT) either as step-and-shot therapy (61% of patients) or by means of volumetric arc therapy (VMAT) in 39% of patients. The MLD, V20_total_lung_, and MED were 12.2 Gy (range: 7–18), 21% (range: 9–35) and 21 Gy (range: 7–34), respectively. All patients received two cycles of induction chemotherapy before radiation therapy followed by immunotherapy with durvalumab in 63/85 (74%) of the cases after completion of RT. Treatment-related parameters are shown in Table 1.

### 3.3. Local Control, Progression Free and Overall Survival

With a median follow-up of 22 months (range: 7.3–66.5), an estimated two-year local control rate of 74% was achieved (see Appendix A Figure A1). The median progression-free and overall survival were 17.3 months (95% CI: 1.9–32.7; see Appendix A Figure A2) and 43.9 months (95% CI: 28.5–59.3; Appendix A Figure A3), respectively. Of the 27/85 (32%) patients who died, 23 cancer-related deaths occurred. The four other patients died from clostridia infection (2) or myocardial infarction (2). The latter were included in the toxicity table, as radiation-induced heart failure could not be entirely excluded (Table 2).

### 3.4. Toxicity

Acute grade 2 and 3 esophagitis was observed in 11/85 (13%) and 7/85 (8%) patients, respectively. In addition, grade 2 and 3 acute pneumonitis was observed in 8/85 (9%) and 3/85 (4%) patients. Two patients experienced this side-effect six months after the end of RT. Because of the long latency to the end of radiation treatment, we assume that in these patients pneumonitis was caused by durvalumab maintenance therapy. DL${}_{\mathrm{CO}}$ and FEV${}_{1}$ were tested in two separate MVA models together with the clinical and therapeutic variables listed in Table 1 for their predictive potential with respect to pneumonitis. PFTs were the only significant predictors for acute pneumonitis, with HRs of 0.696 (95%-CI: 0.492–0.984; *p*-value: 0.040) and 0.278 (95%-CI: 0.101–0.769; *p*-value: 0.014) for DL${}_{\mathrm{CO}}$ and FEV${}_{1}$, respectively. In the observation period of six months after the end of RT, 7/85 (8.2%) patients had pulmonary progress in different constellations: isolated relapse in the lung (2), secondary cancer (1), and intrathoracic recurrence simultaneous with systemic progression (4). Additionally, 8/85 (9.4%) patients experienced disease progression outside the thorax. After exclusion of the seven patients with pulmonary disease progression, the MVA corroborated the results found in the whole cohort: DL${}_{\mathrm{CO}}$ and FEV${}_{1}$ were the only significant predictors for pneumonitis, with HRs of 0.689 (95% CI: 0.477–0.994; *p*-value 0.047) and 0.348 (95% CI: 0.124–0.978; *p*-value 0.045), respectively.

As mentioned above, in 2/85 patients (2%) late side effects in terms of grade 5 heart toxicity could not be completely ruled out. These patients died of heart failure 20 and 17 months after RT, respectively. One of the patients had a history of severe cardiac morbidity prior to irradiation. In this patient, the volume of the heart receiving ≥25 Gy was 12.5%, which was slightly above the threshold of 10% we adopted from QUANTEC [22]. This rather conservative constraint is associated with an estimated 1% probability of cardiac death at 15 years after RT [22]. In the second patient, who had no prior heart disease, the volume of the heart receiving ≥25 Gy was 0%. A summary of radiation-induced toxicity is shown in Table 2.

### 3.5. DL${}_{CO}$ Correlates with V${}_{65-45\%}$

For each patient, the Fischer’s z correlation of the normalized lung density $n_{\rho }$ and corresponding PFT was calculated. Figure 2a shows a patient with a positive correlation between radiographic and PFT changes, while Figure 2b depicts one with inverse correlation.

Several differential volumes revealed a correlation pattern with PFT changes: V${}_{95-85\%}$, V${}_{95-75\%}$, V${}_{95-55\%}$, V${}_{95-25\%}$, V${}_{75-35\%}$, V${}_{65-45\%}$, V${}_{65-35\%}$, and V${}_{65-25\%}$. All of them have in common that the high dose volume between the 95%-isodose and the maximum dose (D${}_{\max}$), i.e., most of the PTV, is excluded. If this high dose area is included, no significant correlation can be found, except for the moderate significance in V${}_{D_{\max}-5\%}$ (Table 3).

Accordingly, the differential volumes that received medium doses revealed the best correlation with DL${}_{\mathrm{CO}}$. Especially, $n_{\rho }$ of V${}_{65-45\%}$ showed positive correlations with FEV${}_{1}$ and DL${}_{\mathrm{CO}}$ in the majority of patients (Figure 3). The dose range of 65–45% of the prescribed dose corresponds to a dose-volume V${}_{42.9--29.7}\;\mathrm{Gy}$ for patients treated with 3 Gy and to V${}_{48.0--33.2}\;\mathrm{Gy}$ for patients treated with 1.8 Gy fractions, respectively.

### 3.6. PFT and CT Density Changes after RT

Figure 4 shows the changes in normalized CT density and PFT values at three and six months after the end of RT relative to baseline. As mentioned above, a decrease in $n_{\rho }$ represents an increase in density. The decline was most significant in $n_{\rho }$ for V${}_{65-45\%}$ from $t_{\mathrm{pre}}$ to $t_{3m}$ as well as from $t_{\mathrm{pre}}$ to $t_{6m}$ (*t*-test, *p*-value < 10${}^{-6}$ ) with an absolute average of −4.8% and −3.9%, respectively. DL${}_{\mathrm{CO}}$ declined moderately from $t_{\mathrm{pre}}$ to $t_{3m}$ (*t*-test, *p*-value = 0.048) by −3.7% on average, while the comparison $t_{\mathrm{pre}}$ to $t_{6m}$ was insignificant (*t*-test, *p*-value = 0.243). FEV${}_{1}$, on the other hand, did not decline significantly between $t_{\mathrm{pre}}$ to $t_{3m}$ (*t*-test, *p*-value = 0.674). The difference between $t_{\mathrm{pre}}$ to $t_{6m}$, however, was moderately significant (*t*-test, *p* = 0.036), with an average decline of −3.2%.

### 3.7. DL${}_{\mathrm{CO}}$ after RT Is Predicted by V${}_{65-45\%}$

Having shown that differential dose volumes in the mid-dose range correlate well with DL${}_{\mathrm{CO}}$, the question arises whether the relative size of a dose volume, i.e., the percentage of lung tissue receiving this dose, can predict changes in DL${}_{\mathrm{CO}}$. In this respect, the highest significance level was again detected for V${}_{65-45\%}$ (Figure 5), which remained significant after Bonferroni correction for multiple testing (Pearson coefficient −0.358, raw *p*-value = 0.003, corrected *p*-value = 0.03; see Appendix A Table A1). A higher percentage of V${}_{65-45\%}$ entailed a lower DL${}_{\mathrm{CO}}$ three months after RT. Likewise, V${}_{75-35\%}$ and V${}_{65-35\%}$ were highly significant, with raw *p*-values of 0.005 and 0.006, respectively (corrected *p*-values = 0.05 and 0.06). Volume ranges including higher and lower dose regions were less significant, e.g., V${}_{95-25\%}$ (raw *p*-value 0.029, corrected *p*-value 0.29) or V${}_{75-25\%}$ (raw *p*-value 0.042, corrected *p*-value 0.42; see Appendix A Table A1). At six months, the correlation between V${}_{65-45\%}$ and DL${}_{\mathrm{CO}}$ was no longer significant (*p*-value = 0.247, see Appendix A Figure A4) as represented by a flatter trend line. A re-calculation after exclusion of the above-mentioned seven patients who experienced pulmonary relapse revealed similar results (Appendix A Figure A5 and Figure A6, Table A2).

## 4. Discussion

In a representative cohort of 85 NSCLC stage III patients comparable to prospective studies [17], the changes in PFT after thoracic RT were moderate. The absolute median decline in FEV${}_{1}$ and DL${}_{\mathrm{CO}}$ was less than 5% within six months after the end of RT, which corroborates previously published results by Grambozov et al. [14] (see Appendix A Figure A4). The fact that the PFTs were the only significant predictors for pneumonitis in MVA underlines the clinical importance of pre-therapeutic PFTs.

The current study showed a significant correlation between PFTs and CT morphology changes after RT. Of note, the best correlation was found for DL${}_{\mathrm{CO}}$ and the differential lung dose volume between the 65%- and 45%-isodose (V${}_{65-45\%}$), which remained significant after Bonferroni correction for multiple testing. In general, the most significant relations (*p* < 0.01) between CT morphology and PFT were found for differential volumes excluding the high-dose area. This finding is not counter-intuitive, as post-therapeutic tumor shrinkage followed by the development of fibrosis is an obviously unpredictable physiological process. This is additionally in line with the clinical practice of excluding the PTV/GTV from the delineation of the lungs as organs at risk in order to safely predict the probability of pneumonitis (e.g., [7,23,24,25]).

Global lung function is represented by PFT parameters such as FEV${}_{1}$ and DL${}_{\mathrm{CO}}$; therefore we hypothesized that regional radiographic density increases, i.e., fibrosis, correlates with PFT decline. Published data on the comparison of PFTs and post-RT fibrosis in the lung are scarce and conflicting. Ma et al., in their cohort of 111 patients, found only a weak correlation between PFTs and CT morphology, with correlation coefficients (CC) between 0.20–0.37 [7]. Interestingly, FEV${}_{1}$ showed higher CCs (0.30–0.37) than DL${}_{\mathrm{CO}}$ (0.17–0.29) [7]. As 91 patients in this cohort were operated on, the higher CC between FEV${}_{1}$ and CT morphology changes can be explained by the fact that a decrease in FEV${}_{1}$ primarily represents a reduction of the airways, which is the case when parts of the lung (including bronchi and bronchioli) are surgically removed.

Ma et al. included patients with only one post-therapeutic measurement, while in our analysis dCTs at three well-defined time points were compared. Additionally, these three CTs were acquired in the same position, which significantly enhanced the registration accuracy compared to Ma et al., who co-registered the pCT with one dCT. Furthermore, in contrast to Ma et al. we analyzed differential volumes, which may be a better approach than considering total lung volume. Finally, we used an individualized patient approach; each patient in our study was his/her own control. These methodological differences may explain why the CCs in our study were almost twice as high as in the investigation by Ma et al. (0.57 for DL${}_{\mathrm{CO}}$ and 0.47 for FEV${}_{1}$). Therefore, we cannot share the viewpoint held by Ma et al. that the validity of PFTs prior to RT is questionable.

The current analysis clearly demonstrated a highly significant correlation of DL${}_{\mathrm{CO}}$ dynamics with the size of a specific radiation volume. Of note, this correlation remained statistically significant after Bonferroni correction for multiple testing (Figure 5), which strongly argues in favor of pre-therapeutic PFTs as a prerequisite for safe thoracic RT. Although this issue is a matter of ongoing debate, our results are in line with other studies in the field (see the review by Niezink et al. [5]).

A retrospective study of 99 patients by Brennan et al. revealed a CC of 0.7 between 4DCT ventilation metrics measured in HU and FEV${}_{1}$ [10], which is on the same order of magnitude as in our study. In this sense, the 4DCT metrics provide additional information on lung function and help with treatment planning to spare critical parts of the lungs in order to retain as much of the vital lung tissue as possible. Major differences of this study compared to ours are the use of 4DCT technology and patient selection with only 60% stage III NSCLC [10].

As mentioned previously, investigations correlating PFTs and CT density in the context of radiation therapy are scarce. Therefore, data on combined pulmonary fibrosis and emphysema (CPFE) may serve as a model. Individuals with this disease have mild airflow limitation combined with a decline in DL${}_{\mathrm{CO}}$, which is more pronounced than in patients with idiopathic pulmonary fibrosis (IPF) or emphysema alone [26]. Both fibrosis and emphysema are areas of low gas exchange; hence, it seems plausible to envisage radiation-induced lung disease as a combination of radiation-induced fibrosis and pre-existing emphysema based on COPD. It is noteworthy that more than half of the patients included in the current analysis had COPD to a certain degree; see Table 1.

The first analysis in the field, conducted by Heremans et al., dates back 30 years [19]. In their 45 patients, the authors described a correlation between FEV${}_{1}$, measured airway obstruction, and CT density changes. As opposed to later investigations which differentiated low (LAA) from high attenuation areas (HAA) [20,21], this pioneering work by Heremans et al. analyzed total lung density [19]. In a study by Matsuoka et al. [20] conducted in 43 CPFE patients, fibrotic and emphysematous areas were defined by HU 0-700 (HAA) and HU <−950 (LAA), respectively. The predictive power of HAA with respect to global lung function measured by DL${}_{\mathrm{CO}}$ was highly significant (*p* < 0.0001). These results were corroborated in two other studies [27,28] including approximately 20 patients each. The authors argue that the physiological explanation for the pronounced decline in DL${}_{\mathrm{CO}}$ could be a reduction in surface area caused by emphysema and fibrosis, which coincides with our results. Therefore, the percentage of HAA, i.e., fibrotic areas, could be a surrogate for global lung function measured by DL${}_{\mathrm{CO}}$.

With due caution, the limited reports available thus far together with the data presented here allow us to assume that a relation between global lung function and regional morphologic changes after RT exists. As that in the current study almost 75% of the patients had a positive correlation between CT density and DL${}_{\mathrm{CO}}$ (Figure 3), it seems that this parameter presents global lung function after RT more accurately than FEV${}_{1}$.

The current study has several obvious limitations. First, PFTs depend on patient compliance and are therefore error-prone. We tried to avoid this bias by measuring lung function at three well defined points of time with each patient as his/her own control. Second, the model-based segmentation algorithm implemented in RayStation^®^ used for contouring of the lungs on each dCT might lead to uncertainties in areas where the tissue adjacent to the lung has a low electron density. Third, the acquisition of the CT data in different patient positions, with or without contrast medium, and on various types of scanners may be a source of errors. In the current study, three dCTs were matched to the pCT. This represents an inherent systematic error characteristic of all studies in the field. Finally, normalization of HU values to the total lung volume could be a problem if denser areas such as the GTV are included for certain points in time and not for others (which, for instance, reflects tumor retraction after treatment). The relative error due to this type of inaccuracy based on tissue changes in the highest dose region (V${}_{D_{\max}-95\%}$) decreases with increasing size of the volume examined (Table 3). The focus of the current analysis, however, was placed on the mid-dose levels.

Despite these shortcomings, a significant correlation between CT density changes and DL${}_{\mathrm{CO}}$ was found.

## 5. Conclusions

The current analysis revealed a strong relation between the dynamics of PFT and CT density changes after RT. The mid-dose range in the lungs, i.e., 65–45% of dose prescribed to the PTV(V${}_{65-45\%}$), was found to be a highly significant predictor for DL${}_{CO}$ after RT. These results demonstrate the importance of routine use of PFTs in the context of curative intent RT, underlining their mutually complementary potential. Although this study was conducted in one of the largest cohorts in the field, prospective studies remain warranted.

## Figures and Tables

**Figure 1 diagnostics-12-01027-f001:**
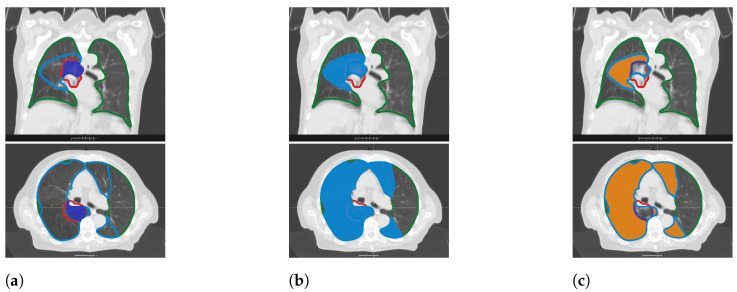
Contours of lungs (green), PTV (red), total dose volumes (blue), and differential dose volumes (orange). (**a**) Contour of total dose volume receiving more than 95% of prescribed dose (V${}_{95\%}$, dark blue). (**b**) Contour of total dose volume receiving more than 15% of prescribed dose (V${}_{15\%}$, light blue). (**c**) Contour of partial dose volume receiving between 95% and 15% of prescribed dose (V${}_{95-15\%}$, orange).

**Figure 2 diagnostics-12-01027-f002:**
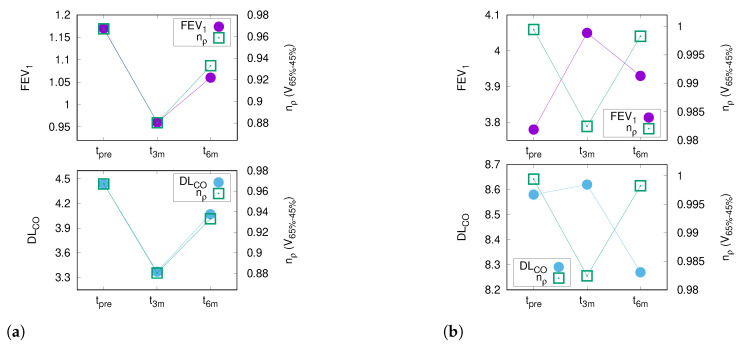
This figure depicts the comparison between the PFT dynamics (circles) and CT density changes (squares). The left vertical axis shows the absolute values for FEV${}_{1}$ or DL${}_{\mathrm{CO}}$, whereas the right y-axis presents the $n_{\rho }$ values for V${}_{65-45\%}$ according to Equation (Equation 2). The horizontal axis depicts the three time points of the PFTs: prior to therapy ($t_{\mathrm{pre}}$) and three ($t_{3m}$) and six months ($t_{6m}$) after the end of RT. The correlation between the differential volume V${}_{65-45\%}$ and FEV${}_{1}$ is shown in the top row, while the bottom row depicts the relation with DL${}_{\mathrm{CO}}$. The left column (**a**) presents the data for a patient with a “positive” correlation: the decline of FEV${}_{1}$ and DL${}_{\mathrm{CO}}$ three months after the end of RT ($t_{3m}$) is mirrored by an increase in CT density, which almost fully recovers three months later. The right column (**b**) shows a “negative” relation: the improvement of FEV${}_{1}$ and DL${}_{\mathrm{CO}}$ is accompanied by an increase in CT density.

**Figure 3 diagnostics-12-01027-f003:**
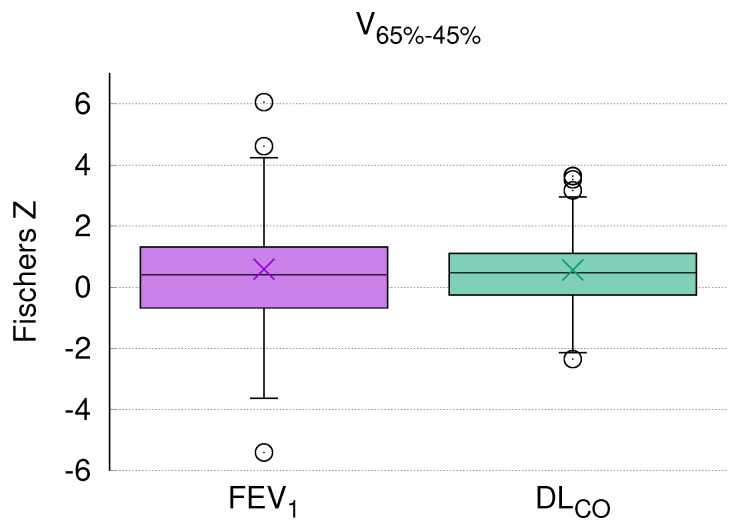
Fischers Z correlation coefficients calculated individually for each patient between $n_{\rho }$ of V${}_{65-45\%}$ and the respective PFT parameter (FEV${}_{1}$ and DL${}_{\mathrm{CO}}$). Lines in boxplots indicate the median, x indicates the mean value, and whiskers are set to 1.5 times the interquartile range. Almost 75% of the patients had a positive correlation between morphologic changes and DL${}_{\mathrm{CO}}$ (green box).

**Figure 4 diagnostics-12-01027-f004:**
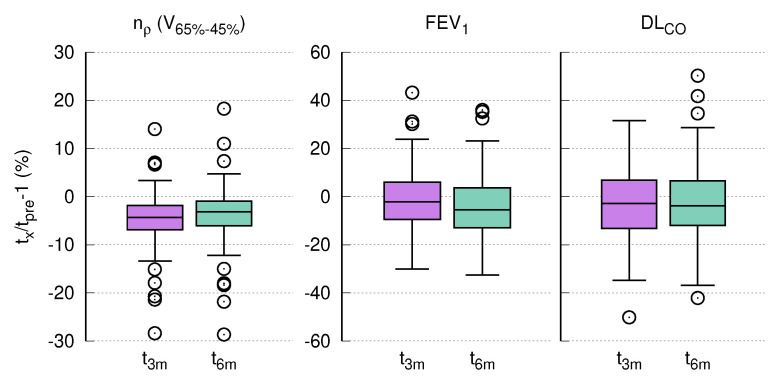
Relative temporal development of normalized CT density $n_{\rho }$ and PFTs for three months ($t_{3m}$) and six months ($t_{6m}$) after treatment using three months pre-treatment ($t_{\mathrm{pre}}$) as a baseline.

**Figure 5 diagnostics-12-01027-f005:**
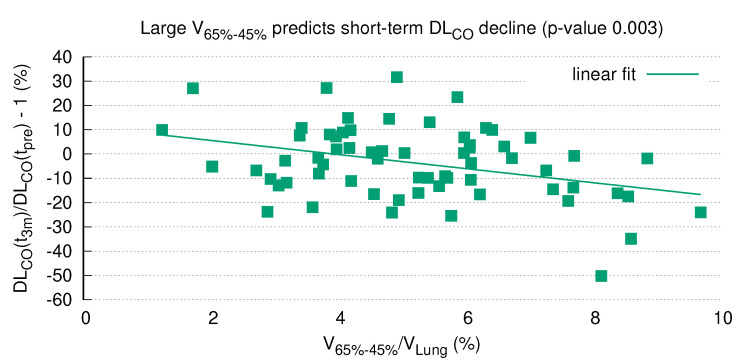
DL${}_{\mathrm{CO}}$ three months after RT (Y-axis) is correlated with the relative size of the differential volume V${}_{65-45\%}$ (X-axis). Negative values on the Y-axis represent a decline of DL${}_{\mathrm{CO}}$ compared to baseline. V${}_{65-45\%}$ correlates significantly with DL${}_{\mathrm{CO}}$ decline three months after RT (Pearson correlation *p*-value 0.003; Bonferroni correction for multiple testing *p*-value 0.03); the higher the proportion of V${}_{65-45\%}$, the lower the DL${}_{\mathrm{CO}}$.

**Table 1 diagnostics-12-01027-t001:** Patient and treatment characteristics. ECOG = Eastern Cooperative Oncology Group, RT = radiotherapy, CCI = Charlson Comorbidity Index, MED = mean esophageal dose, EQD${}_{2}$ = biologically equivalent dose in 2 Gy fractions, MLD 0 mean lung dose, CTX = chemotherapy, IO = immunotherapy, GTV = gross tumor volume, DLCOc = corrected carbon monoxide diffusion capacity, V20_total_lung_ = volume of the lungs receiving at least 20 Gy, NSCLC = non-small cell lung cancer. Weight loss within six months before diagnosis was considered.

Patients N = 85		
Age (years)	median	66
	range	46–81
Sex	male	59 (69%)
	female	26 (31%)
Weight loss (%)	>5%	9 (11%)
	<5%	76 (89%)
ECOG	0–1	83 (98%)
	2	2 (2%)
Smoking status	ex	47 (55%)
	current	27 (32%
	never	9 (11%)
	unknown	2 (2%)
Histology	NSCLC	85 (100%)
	unknown	0 (0%)
N-stage	0	2 (2%)
	1	7 (8%)
	2	57 (68%)
	3	18 (22%)
UICC	III	85 (100%)
FEV${}_{1}$ (L)	median	2.2
	range	0.8–3.8
DLCOc (mmol/min*kPa)	median	5.3
	range	1.4–11.0
COPD grade	0	46 (54%)
	1	6 (7%)
	2	18 (21%)
	3	12 (14%)
	4	4 (5%)
	unknown	0 (0%)
CCI	median	5
	range	2–9
**Treatment**		
GTV (ml)	median	19
	range	0.3–308
Tumor location (n)	peripheral	44 (52%)
	central	41 (48%)
RT technique (n)	IMRT	52 (61%)
	VMAT	33 (39%)
Systemic therapy (n)	CTX	22 (26%)
	CTX + IO	63 (74%)
MLD (Gy)	median	12.2
	range	7–18
V20_total_lung_ (%)	median	21%
	range	9–35%
MED (Gy)	median	21
	range	7.5–34
EQD${}_{2}$ (Gy)	median	72.3
	range	58.3–88.2

**Table 2 diagnostics-12-01027-t002:** Treatment-related toxicity (n.a. = not assessed).

Toxicity (N = 85)						
Type of toxicity		Grade 1	Grade 2	Grade 3	Grade 4	Grade 5
Acute	Esophagus	n.a.	11 (13%)	7 (8%)	0	0
	Lung	n.a.	8 (9%)	3 (4%)	0	0
Late	Esophagus	n.a.	0	0	0	0
	Lung	n.a.	0	0	0	0
	Heart	n.a.	0	0	0	2 (2%)

**Table 3 diagnostics-12-01027-t003:** Overview of confidence levels for the correlation between post-therapeutic dynamics in radiographic alterations within the differential volumes (higher dose–lower dose) and PFT dynamics. This correlation is represented for FEV${}_{1}$ (left) and DL${}_{\mathrm{CO}}$ (right), respectively. High and moderate confidence levels are shown in blue (*p* < 0.01), green (*p* < 0.05) and red (not significant).

Isodose		Lower Dose																			
Volume		95%		85%		75%		65%		55%		45%		35%		25%		15%		5%	
higher dose	max																				
	95%																				
	85%																				
	75%																				
	65%																				

## Data Availability

This study did not report any data.

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
