# Peer review of "Carbon Monoxide Diffusing Capacity (DLCO) Correlates with CT Morphology after Chemo-Radio-Immunotherapy for Non-Small Cell Lung Cancer Stage III"

_diagnostics, 2022, doi:10.3390/diagnostics12051027_

Round 1
Reviewer 1 Report
The authors try to develop a model for prediction of PF and DLCO changes after radio therapy in nsclc III patients. V65%−45% show good correlation.
We think the article offers some valuble new insights in radiomics ands its correlations with pulmonal function.
But we have some major and minor qustions we like the authors to answer before publication:
Major
- Where there no patients receiving simultaneous chemoradiotherapy (line 167)?
- As I understand patients were treated with different dose prescriptions (median EQD2 of 72.3 Gy (58.3-88.2)). To my understanding it would be more feasible to use Volumes for 20, 40, 50 Gy and not V95%−85%. of the dose applied.
- How many patients had progress in the lung during the time of observation and how does this affect PFT, were there patients with IO induced pneumonitis?
Minor:
- Table 1:
- Histology unknow (80%) - percentage unclear
- UICC N-status?
- DLCOC? (instead of DLCO)
- PTV (50 Gy > 700ccm negative predictor for survival)
- EQD2 dose very high, we expect max of 66Gy (please comment on high prescribed doses e.g. 66Gy in 3 Gy daily fractions)
- Appendix F:
- X and y axes has to start with 0
- Intervals for time in 6 or 12 month
Author Response
The authors try to develop a model for prediction of PF and DLCO changes after radio therapy in nsclc III patients. V65%−45% show good correlation. We think the article offers some valuble new insights in radiomics ands its correlations with pulmonal function. But we have some major and minor qustions we like the authors to answer before publication:
Major
- Where there no patients receiving simultaneous chemoradiotherapy (line 167)?
- As I understand patients were treated with different dose prescriptions (median EQD2of 72.3 Gy (58.3-88.2)). To my understanding it would be more feasible to use Volumes for 20, 40, 50 Gy and not V95%−85%. of the dose applied.
- How many patients had progress in the lung during the time of observation and how does this affect PFT, were there patients with IO induced pneumonitis?
Minor:
- Table 1:
- Histology unknow (80%) - percentage unclear
- UICC N-status?
- DLCOC? (instead of DLCO)
- PTV (50 Gy > 700ccm negative predictor for survival)
- EQD2 dose very high, we expect max of 66Gy (please comment on high prescribed doses e.g. 66Gy in 3 Gy daily fractions)
- Appendix F:
- X and y axes has to start with 0
- Intervals for time in 6 or 12 month
Submission Date
23 February 2022
Date of this review
01 Mar 2022 17:34:59
We thank reviewer 1 for his / her valuable comments, please find our point-by-point responses below. Insertions in the manuscript text are in blue font.
- Where there no patients receiving simultaneous chemoradiotherapy (line 167)?
No. Our department follows a historically grown tradition of sequential high-dose irradiation based on the organisational and institutional surroundings with pneumologists and medical oncologists referring from small peripheral hospitals. After two cycles of induction chemotherapy patients receive sequential high-dose irradiation to make up for the delayed start of radiation treatment.
- As I understand patients were treated with different dose prescriptions (median EQD2of 72.3 Gy (58.3-88.2)). To my understanding it would be more feasible to use Volumes for 20, 40, 50 Gy and not V95%−85%.of the dose applied.
We thank reviewer 1 for bringing up this important issue, which contributes substantially to a better understanding of the paper. First of all we want to clarify that the starting point of our analysis was the question, in how far do radiomorphologic changes in postRT CT scans correlate with pulmonary function changes. In other words: is there a correlation between the dynamics of HU changes (representing radiomorphologic changes) and PFT parameters. If one looks at the mid-dose ranges (excluding the tumor, i.e. Dmax – 95% isodose) the correlation between HU changes in this area and the global lung function is very close. The results are summarized in table 3 and in lines 204 to 208. From the various mid-dose range volumes which showed a good correlation with HU changes (table 3), V65%-45% had the best correlation with DLCO after treatment, which is a pertinent finding of the current analysis. Although it may be true that V20, V40 and V50 – as suggested by the reviewer – are more common DVH parameters that may correlate with PFTs they do not necessarily correlate with HU changes.
In particular, the dose region denoted by V20Gy would – in our notation – be VDmax-30,3% for a prescribed dose of 66 Gy and VDmax-27,1% for a prescribed dose of 73,8 Gy. As can be seen from Table 1, there is no correlation between either PFT parameter with VDmax-25%. The same is true for V40Gy (= VDmax-30,6% and VDmax-54,2%, respectively) and for V50Gy (= VDmax-75,6% and VDmax-67,8,6%, respectively). The reason for this may be the fact that V20, V40 and V50 include the high-dose region (Dmax-V95%), in which the post-therapeutic tissue changes are hardly predictable due to tumor shrinkage and the generation of fibrosis (Discussion, end of the first paragraph).
In order to address the reviewer’s query, the correlations between the size of the suggested volumes and DLCO are shown below for illustration purposes. Not surprisingly, the correlations are significant. But – as opposed to V65%-45% - the relation with HU dynamics is insignificant. Therefore these conventionally used volumes are less versatile since radiographic changes are not mirrored in these volumes. Hence they do not complement PFTs and cannot be used as potential surrogates for global lung function.
- How many patients had progress in the lung during the time of observation and how does this affect PFT, were there patients with IO induced pneumonitis?
Progress in the lung
In the observation period of 6 months after the end of RT 7/85 (8.2%) patients had pulmonary progress in different constellations: isolated relapse in the lung (2), secondary cancer (1) and intrathoracic recurrence simultaneously with systemic progression (4). Additionally, 8/85 (9.4%) patients experienced disease progression outside the thorax.
Based on the reviewer’s input, we re-calculated the correlations between partial volumes and PFTs excluding the patients with pulmonary progression, which corroborated the results in the whole cohort. Therefore we added the table and figures below in supplementary material (figure 5, supplementary table 4, supplementary figure A9). On top of that, the multivariate analysis by means of Cox regression was also re-done in this smaller patient cohort (p-value 0.047; HR = 0.689; 95%-CI 0.477 – 0.994). The manuscript was modified as follows in results sections 3.4 Toxicity and 3.7 DLCO after RT is predicted by V65%-45%.
In the observation period of six months after the end of RT 7/85 (8.2%) patients had pulmonary progress in different constellations: isolated relapse in the lung (2), secondary cancer (1) and intrathoracic recurrence simultaneously with systemic progression (4). Additionally, 8/85 (9.4%) patients experienced disease progression outside the thorax. After exclusion of the seven patients with pulmonary disease progression the MVA corroborated the results found in the whole cohort: DLCO and FEV1 were the only significant predictors with HRs of 0.689 (95% CI: 0.477 – 0.994; p-value 0.047) and 0.348 (95% CI: 0.124 – 0.978; p-value 0.045), respectively.
(…)
A re-calculation after exclusion of the above mentioned seven patients who experienced pulmonary relapse revealed similar results (supplementary Figures A10 and A11, Table 5)
Figure A 10. DLCO at three months after RT (y-axis) is correlated to the relative size of the differential volume V65%-45% (x-axis): negative values on the y-axis represent a decline of DLCO compared to baseline. After exclusion of the seven patients with pulmonary progress, V65%-45% still correlates significantly with DLCO decline three months after thoracic RT (Pearson correlation, p-value 0.008): the higher the proportion of V65%-45% the lower DLCO.
Table 5 Table of Pearson correlations between differential volumes and the ratio of DLCO t3m/tpre excluding seven patients with pulmonary progress. In this correlation testing only those volumes were included, whose dynamics showed a significant correlation with PFT (either FEV1 or DLCO) Table 3.
Figure A 11. Cohort excluding seven patients with pulmonary progression (N = 78). At six months the correlation between the relative size of V65%-45% and the decline in DLCO is no longer significant, which is visualized by a flatter trend line (Pearson correlation, p-value = 0.349).
IO induced pneumonitis
There were 11/85 (13%) patients with clinically relevant pneumonitis. Two patients experienced this side-effect at six months after the end of RT. Thus in these two cases it can be assumed that it was induced by durvalumab maintenance therapy. In the other nine cases it was most likely caused by radiation treatment or a combination of the two modalities.
Two patients experienced this side-effect six months after the end of RT. Because of the long latency to the end of radiation treatment, we assume that in these patients pneumonitis was caused by durvalumab maintenance therapy.
- Table 1
According to the reviewer’s suggestions we modified table 1, which can be found below (changes in blue font). We apologize for the printing errors regarding DLCOc and histology (should be 0% instead of 80%). N-stage was added to the patient table. The maximum PTV was 545 ccm so that the 700ccm cutoff – as mentioned by reviewer 1 – was not reached in our cohort. With respect to EQD2 the following passage was inserted into the results section 3.2 (also see issue 1 above):
Sixty-four patients received dose-differentiated accelerated radiotherapy (DART) as described previously (Grambozov 2019) with total radiation doses between 73.8 and 90 Gy depending on tumor size. As of 2020/01 patients received 66 Gy in 3 Gy fractions, which is biologically in the same range as 73.8 Gy in 1.8 Gy fractions.
Table 1: Patient and treatment characteristics. ECOG = Eastern Cooperative Oncology Group, FEV1 = forced expiratory volume in 1 second, RT = radiotherapy, CCI = Charlson Comorbidity Index, MED = mean esophageal dose, EQD2 = biologically equivalent dose in 2 Gy fractions, MLD mean lung dose, CTX = chemotherapy, IO = immunotherapy, GTV = gross tumor volume, DLCOc = corrected carbon monoxide diffusing capacity, V20total lung = volume of the lung receiving at least 20 Gy, NSCLC = non-small cell lung cancer. 1Weight loss within six months before diagnosis was considered.
- Appendix F
We adjusted the intervals in the figures as requested. The starting point for the scales (= 0) on x- and y-axis is default in the commercially available SPSS version 27.
Figure A6. The 2-year local control rate was 74%.
Figure A7. The median progression-free survival was 17.3 months (95% CI: 1.9 – 32.7).
Figure A8: The median overall survival was 43.9 months (95% CI: 28.5 – 59.3).

Reviewer 2 Report
The paper present new and important data of interest to general medical, oncology, pulmonology and radiation treatment generally. The authors studied the relationship between CT changes and diffusion capacity changes, and PFT after RT for NSCLC. Language use is generally good, but is frequently a bit careless, as outlined below. The entire paper must be edited and proofread. It appears this has not been done previously. An important paper for non-specialists like GPs and general oncologist should be readily understandable to them. May I suggest that the authors ask their general medical or general oncology colleagues to look over the ms. and suggest changes to allow easier reading ? Asking an early career junior Dr. [as a first year after graduation Dr-in-training] to help make their ms. clearer and offering him/her coauthorship would have prevented much of the silly errors and unclear writing in this ms.
Listed "Keywords" are not keywords, they are mostly abbreviations. A seperate heading Abbreviations is needed. Also many abbreviations used in text are not listed.
Titles should use only the most well known abbreviations [DNA, CT or MRI scan, HIV, etc]. Abbreviations in Abstract must be few and preferably none. If any used in Abstract they must be defined in the Abstract.
Spell out "carbon monoxide diffusing capacity" in title.
Use of two terms for same thing should be clarified. So, "Pulmonary function tests (PFTs, spirometry)" would reduce chance of unclarity. Better might be to use only PFT ?
Pg 2 line 37, comma should be full stop.
Pg 2 line 39, again, dont use 2 terms for same thing. CT or CT scan are either acceptable to use without specifying but dont use both. You could say "...computed tomography (CT)..." if you wanted, and thereafter use only CT.
Pg 2 line 47, DLCO was not previously defined.
Pg 2 line 48, use of "FEV1" is example of an abbreviation common enough in all fields of medicine as to not require spelling out in words. I would include CT in this category as well.
Pg 2 line 48, again, use simply "CT densities"
Pg 2 line 49, should read "Ma et al". Throughout ms. et al must be added in reference to papers with many authors.
Pg 2 line 51, sorry I can't understand what "radiobiological notion that the lung is a parallel organ, radiation induced fibrosis would correlate with global lung function changes." means. That PFT and CT density should correlate is easy to understand and is a correlation we would guess to be present.
I recommend not using RILD abbreviation. Authors use this rarely and fewer abbreviations better.
Pg 2 line 56, again reflecting the authors' carelessness in writing of their report, RILD is defined in line 2 pg 1 then again line 56 pg 2.
Is it too picky to complain that the authors might consider that CT density and fibrosis are not identical entities ? We assume they are in irradiation induced context, but in an academic paper maybe the distinction, or potential distinction should be mentioned ?
Pg 2 line 59, would it not be more correct to say "with curative intent" ? We expect some folks will have reccurrence.
In Section 2.1. The authors don't mention that they studied only Stage III pts. Correct ?
I will not comment on the careless writing after pg 2. It continues frequently thereafter however.
Use of standard cancer chemotherapy agents like carboplatin, etoposide etc need not be introduced but newer agents like must be introduced with a short sentence or two. For example "Durvalumab (Imfinzi ™) is a programmed death-ligand 1 (PD-L1) blocking antibody for use in adjunctive treatment of…". Also note that generic names of drugs start with lower case letter unless at beginning of sentence. Proprietary names always start with upper case.
Pg 3 Line 94, is not bronchiolar narrowing also reflected in reduced FEV1 ?
Pg 3 line 103 "HU histogram" not previously defined.
Again line 113, ROI is common enough in general medical education as to not require prior definition as to what the abbreviation stands for.
Again showing the utter carelessness of the writing of this manuscript HU is defined on line 122 after its first use on line 103. Then they define it again in line 128 ! Have none of the authors proofread their manuscript ?
Line 157, use of "COPD"here is ok.
Line 169, use durvalumab with lower case d.
In Table 1, I was unclear over what time interval the weight loss occurred. All abbreviations in a Table must be defined in the legend. They are not in this paper.
Figure on top pg 7 is intolerably disorganized.
Fig. 3 is of particular value to our understanding of post irradiation lung changes.
Pg 9 line 263, the fault could well be mine, but I didn't understand "As FEV1 was introduced into the radio-oncological literature based on surgery data this finding may reflect the fact that 91 patients in this cohort were operated."
Line 273 is a strong point. This matter would best be in its own paragraph. The authors are arguing a clinical practice matter, and correctly in my opinion, showing that PFT is required prior to RT. Authors, please make this point clearer and in its own separate paragraph. Also restate your conclusion in the Abstract.
Author Response
The paper present new and important data of interest to general medical, oncology, pulmonology and radiation treatment generally. The authors studied the relationship between CT changes and diffusion capacity changes, and PFT after RT for NSCLC. Language use is generally good, but is frequently a bit careless, as outlined below. The entire paper must be edited and proofread. It appears this has not been done previously. An important paper for non-specialists like GPs and general oncologist should be readily understandable to them. May I suggest that the authors ask their general medical or general oncology colleagues to look over the ms. and suggest changes to allow easier reading ? Asking an early career junior Dr. [as a first year after graduation Dr-in-training] to help make their ms. clearer and offering him/her coauthorship would have prevented much of the silly errors and unclear writing in this ms.
Listed "Keywords" are not keywords, they are mostly abbreviations. A seperate heading Abbreviations is needed. Also many abbreviations used in text are not listed.
Titles should use only the most well known abbreviations [DNA, CT or MRI scan, HIV, etc]. Abbreviations in Abstract must be few and preferably none. If any used in Abstract they must be defined in the Abstract.
Spell out "carbon monoxide diffusing capacity" in title.
Use of two terms for same thing should be clarified. So, "Pulmonary function tests (PFTs, spirometry)" would reduce chance of unclarity. Better might be to use only PFT ?
Pg 2 line 37, comma should be full stop.
Pg 2 line 39, again, dont use 2 terms for same thing. CT or CT scan are either acceptable to use without specifying but dont use both. You could say "...computed tomography (CT)..." if you wanted, and thereafter use only CT.
Pg 2 line 47, DLCO was not previously defined.
Pg 2 line 48, use of "FEV1" is example of an abbreviation common enough in all fields of medicine as to not require spelling out in words. I would include CT in this category as well.
Pg 2 line 48, again, use simply "CT densities"
Pg 2 line 49, should read "Ma et al". Throughout ms. et al must be added in reference to papers with many authors.
Pg 2 line 51, sorry I can't understand what "radiobiological notion that the lung is a parallel organ, radiation induced fibrosis would correlate with global lung function changes." means. That PFT and CT density should correlate is easy to understand and is a correlation we would guess to be present.
I recommend not using RILD abbreviation. Authors use this rarely and fewer abbreviations better.
Pg 2 line 56, again reflecting the authors' carelessness in writing of their report, RILD is defined in line 2 pg 1 then again line 56 pg 2.
Is it too picky to complain that the authors might consider that CT density and fibrosis are not identical entities ? We assume they are in irradiation induced context, but in an academic paper maybe the distinction, or potential distinction should be mentioned ?
Pg 2 line 59, would it not be more correct to say "with curative intent" ? We expect some folks will have reccurrence.
In Section 2.1. The authors don't mention that they studied only Stage III pts. Correct ?
I will not comment on the careless writing after pg 2. It continues frequently thereafter however.
Use of standard cancer chemotherapy agents like carboplatin, etoposide etc need not be introduced but newer agents like must be introduced with a short sentence or two. For example "Durvalumab (Imfinzi ™) is a programmed death-ligand 1 (PD-L1) blocking antibody for use in adjunctive treatment of…". Also note that generic names of drugs start with lower case letter unless at beginning of sentence. Proprietary names always start with upper case.
Pg 3 Line 94, is not bronchiolar narrowing also reflected in reduced FEV1 ?
Pg 3 line 103 "HU histogram" not previously defined.
Again line 113, ROI is common enough in general medical education as to not require prior definition as to what the abbreviation stands for.
Again showing the utter carelessness of the writing of this manuscript HU is defined on line 122 after its first use on line 103. Then they define it again in line 128 ! Have none of the authors proofread their manuscript ?
Line 157, use of "COPD"here is ok.
Line 169, use durvalumab with lower case d.
In Table 1, I was unclear over what time interval the weight loss occurred. All abbreviations in a Table must be defined in the legend. They are not in this paper.
Figure on top pg 7 is intolerably disorganized.
Fig. 3 is of particular value to our understanding of post irradiation lung changes.
Pg 9 line 263, the fault could well be mine, but I didn't understand "As FEV1 was introduced into the radio-oncological literature based on surgery data this finding may reflect the fact that 91 patients in this cohort were operated."
Line 273 is a strong point. This matter would best be in its own paragraph. The authors are arguing a clinical practice matter, and correctly in my opinion, showing that PFT is required prior to RT. Authors, please make this point clearer and in its own separate paragraph. Also restate your conclusion in the Abstract.
We thank reviewer 2 for his/her thorough evaluation of our manuscript. Please find our point-by-point answers below (insertions in the manuscript text in blue font).
- Listed "Keywords" are not keywords, they are mostly abbreviations. A seperate heading Abbreviations is needed. Also many abbreviations used in text are not listed. The list of keywords was modified as follows and the list of abbreviations completed. The heading “Abbreviations“ was already included in the first submission at the end of the original manuscript (after the conflict of interest statement). In the current version of the manuscript the list of abbreviations was updated.
Non-small cell lung cancer, dose volume histogram, carbon monoxide diffusing capacity, high dose radiation, radiation induced lung disease
- Titles should use only the most well known abbreviations [DNA, CT or MRI scan, HIV, etc]. Abbreviations in Abstract must be few and preferably none. If any used in Abstract they must be defined in the Abstract. Title and abstract were adjusted accordingly.
Title: Carbon monoxide diffusing capacity (DLco) correlates with CT morphology after chemo-radio-immunotherapy for non-small cell lung cancer stage III.
Abstract, line 8: Eighty-five patients with non-small cell lung cancer stage III (NSCLC) treated between (...) ; line 11-12: Pulmonary function tests (PFTs) were performed three as well as six months after completion of radiotherapy (RT) and compared to baseline. Line 15: Differential volumes defined by specific isodoses were generated to correlate them with the PFTs. Line 18: V65%-45% was also a predictor for DLCO after RT (Pearson correlation coefficient -0.358; corrected p-value 0.03). Line 19: In multivariate analysis (Cox-Regression) DLCO (HR 0.696; p-value 0.040) and FEV1 (HR 0.278; p-value 0.014) predicted pneumonitis. Line 22 (also see issue 29 below): The current analysis revealed a strong relation between the dynamics of DLCO and CT morphology changes in the mid-dose range, which is a strong indicator for the importance of routinely used PFTs in the context of curatively intended treatment.
- Spell out "carbon monoxide diffusing capacity" in title. Please also see above (point 2)
Title: Carbon monoxide diffusing capacity (DLco) correlates with CT morphology after chemo-radio-immunotherapy for non-small cell lung cancer stage III.
- Use of two terms for same thing should be clarified. So, "Pulmonary function tests (PFTs, spirometry)" would reduce chance of unclarity. Better might be to use only PFT? As suggested by the reviewer we decided to use PFT (= pulmonary function test) throughout the manuscript and deleted “spirometry“, which appeared once in the whole text (line 36) so that this sentence now runs as follows:
Set up inaccuracies and patient compliance may bias PFTs so that a reproducibility within a +/- 10% margin was reported (Borst, G.R.).Nevertheless, …
- Pg 2 line 37, comma should be full stop. We corrected this error (please also see issue 4).
Set up inaccuracies and patient compliance may bias PFTs so that a reproducibility within a +/- 10% margin was reported (Borst, G.R.).Nevertheless, …
- Pg 2 line 39, again, dont use 2 terms for same thing. CT or CT scan are either acceptable to use without specifying but dont use both. You could say "...computed tomography (CT)..." if you wanted, and thereafter use only CT. As suggested by the reviewer, the acronym “CT“ for computed tomography is used throughout the manuscript now.
Abstract, line 4: We hypothesized that the decrease in global PF corresponds to the increase in tissue density in follow-up CT. Line 13: At the same time points patients had a diagnostic CT (dCT).
Line 39: When comparing the density values of CT acquired (…)
- Pg 2 line 47, DLCO was not previously defined. We corrected this error.
Line 47: The aim of this retrospective study was to correlate the dynamics of carbon monoxide diffusing capacity (DLCO) and FEV1, which are the most frequently used lung function parameters in clinic, with that of CT densities.
- Pg 2 line 48, use of "FEV1" is example of an abbreviation common enough in all fields of medicine as to not require spelling out in words. I would include CT in this category as well. We agree with the reviewer. Therefore, the redundant definitions in line 92 and the abbreviations section were deleted.
- Pg 2 line 48, again, use simply "CT densities". The sentence was modified according to the reviewer’s suggestion (please also see above issue 7).
Line 47-49: The aim of this retrospective study was to correlate the dynamics of carbon monoxide diffusing capacity (DLCO) and FEV1, which are the most frequently used lung function parameters in clinic, with that of CT densities.
- Pg 2 line 49, should read "Ma et al". Throughout ms. et al must be added in reference to papers with many authors. Whenever a reference with multiple authors is quoted “et al.“ was added to the name of the first author throughout the manuscript.
Line 49: In accordance with Ma et al. (…)
Please also see lines 267, 268, 272, 273, 290, 294, 303
- Pg 2 line 51, sorry I can't understand what "radiobiological notion that the lung is a parallel organ, radiation induced fibrosis would correlate with global lung function changes." means. That PFT and CT density should correlate is easy to understand and is a correlation we would guess to be present. The correlation between PFT and CT density changes is not always straightforward. Clinical practice shows that every once in a while patients who develop fibrosis after radiation treatment do not have a measurable impairment of their lung function, and vice versa. An example of a patient with a “negative correlation“ is presented in figure 2b. In this patient the PFT three months after RT improves, while CT density increases.
As for the “radiobiological notion …“, we admit that in the current context this is hard to understand and a lengthy explanation of the difference between parallel and serial organs is beyond the scope of the current manuscript. In brief, what is meant here, is the fact that in parallel organs like the lungs a very high dose to a small volume does not impact the physiological functioning of the organ. However, if this dose exceeds a certain critical volume global lung function is impaired. In order to avoid misunderstandings we left this concept out and changed the respective passages in the manuscript accordingly.
Line 51: In accordance with Ma et al. [7], we hypothesized that a decline in PFT would correlate to an increase in lung tissue density.
(…)
Discussion, line 254: Global lung function is represented by …
Discussion, lines 309-311: Although firm conclusions cannot be drawn, the mechanism could be similar to CPFE (Matsuoka, Gomes).
Abstract, line 2: We hypothesized that the decrease (…)
- I recommend not using RILD abbreviation. Authors use this rarely and fewer abbreviations better. As suggested, we do not use this acronym and also deleted it from the abbreviations section.
- Pg 2 line 56, again reflecting the authors' carelessness in writing of their report, RILD is defined in line 2 pg 1 then again line 56 pg 2. We apologize for being inaccurate – please see above, issue 12.
- Is it too picky to complain that the authors might consider that CT density and fibrosis are not identical entities ? We assume they are in irradiation induced context, but in an academic paper maybe the distinction, or potential distinction should be mentioned ? In order to clarify this, we inserted the following sentence in the first paragraph of section 2.4 CT morphology changes.
Although one has to be aware that CT density and fibrosis are two different entities, in the context of radiation treatment however, it is plausible to assume that the anatomical substrate of CT density increases is lung tissue fibrosis.
- Pg 2 line 59, would it not be more correct to say "with curative intent" ? We expect some folks will have reccurrence. As suggested we modified the sentence.
Line 59: Eighty-five patients who underwent thoracic radiation treatment with curative intent between October 2015 and October 2020 were included.
- In Section 2.1. The authors don't mention that they studied only Stage III pts. Correct? We added this information at the beginning of section 2.1 Patients.
Lines 62 – 63: The tumor was histologically or cytologically verified and categorized as stage III according to the 8th edition of the TNM classification.
- I will not comment on the careless writing after pg 2. It continues frequently thereafter however. We apologize for linguistic and conceptual inaccuracies, which we corrected by thorough proof reading. Changes and corrections in the manuscript texts were inserted in the lines indicated below (the numbers refer to the original version). A list of changes also appears at the end of the current version of the manuscript. We hope that the reviewer will be satisfied with the current version of the manuscript.
Lines: 66, 69, 70, 78-79, 101-102, 119, 126, 137, 156, 158, 164, 165, 167, 172, 185-189, 199-201, 207, 209-10, 212, 216-21, 223, 239, 243-45, 249, 253-58, 264-65, 267-68, 272, 281-82, 286-87, 290-92, 294, 301, 303-06, 309-12, 318-23, 325, 330, 339-41
- Use of standard cancer chemotherapy agents like carboplatin, etoposide etc need not be introduced but newer agents like must be introduced with a short sentence or two. For example "Durvalumab (Imfinzi ™) is a programmed death-ligand 1 (PD-L1) blocking antibody for use in adjunctive treatment of…". Also note that generic names of drugs start with lower case letter unless at beginning of sentence. Proprietary names always start with upper case. Following the reviewer’s suggestion we inserted the following sentence at the end of section 2.2 Treatments: chemotherapy, irradiation, immune checkpoint inhibition.
Third, as of September 2017, patients received durvalumab (ImfinziÒ) 10 mg/kg maintenance therapy for one year after the end of RT(Antonia S.J. 2018). Durvalumab is a monoclonal antibody that blocks programmed death ligand 1 (PD-L1) thereby enabling T cells to better recognize the tumor. It is used in the adjuvant treatment of stage III NSCLC after chemo-radiotherapy.
- Pg 3 Line 94, is not bronchiolar narrowing also reflected in reduced FEV1 ? We modified the sentence as follows:
(…) for the narrowing of large or medium-sized bronchi as well as the bronchiolar airways, while (…)
- Pg 3 line 103 "HU histogram" not previously defined. We apologize for the error.
Line 103: The body outline was contoured based on the Hounsfield unit (HU) histogram (i.e. Gray Level Threshold in RayStationÒ) for each dCT dataset.
- Again line 113, ROI is common enough in general medical education as to not require prior definition as to what the abbreviation stands for. We agree.
- Again showing the utter carelessness of the writing of this manuscript HU is defined on line 122 after its first use on line 103. Then they define it again in line 128 ! Have none of the authors proofread their manuscript ? Again, we apologize for the inaccuracy. In the current version, the term is defined in line 103 (please also see issue 20, above) and its abbreviation is used throughout the text.
Line 103: The body outline was contoured based on the Hounsfield unit (HU) histogram (i.e. Gray Level Threshold in Raystation™) for each dCT dataset.
- Line 157, use of "COPD"here is ok. Thank you.
- Line 169, use durvalumab with lower case d. We changed it accordingly.
- In Table 1, I was unclear over what time interval the weight loss occurred. All abbreviations in a Table must be defined in the legend. They are not in this paper. We inserted a number in the third line “Weight loss (%)1“, which is explained in the legend. Please find the modified table and legend below.
Table 1: Patient and treatment characteristics. ECOG = Eastern Cooperative Oncology Group, FEV1 = forced expiratory volume in 1 second, RT = radiotherapy, CCI = Charlson co-morbidity index, MED = mean esophageal dose, EQD2 = biologically equivalent dose in 2 Gy fractions, MLD mean lung dose, CTX = chemotherapy, IO = immunotherapy, GTV = gross tumor volume, DLCOc = corrected carbon monoxide diffusing capacity, V20total lung = volume of the lung receiving 20 Gy or more, NSCLC = non-small cell lung cancer. 1Weight loss within six months before diagnosis was considered.
- Figure on top pg 7 is intolerably disorganized. We modified the figure as well as the caption and sincerely hope that the reviewer may find the current version acceptable.
|
(a) Patient with a positive correlation between PFT and CT density dynamics. |
(b) Patient with a negative correlation between PFT and CT density dynamics. |
This figure depicts the comparison between PFT dynamics (circles) and CT density changes (squares). The left vertical axis shows the absolute values for FEV1 or DLCO, whereas the right vertical axis presents the nρ values for V65%-45% according to equation 2. The horizontal axis depicts the three time points of the PFTs: prior to therapy (tpre), three (t3m) and six months (t6m) after the end of RT. The correlation between the differential volume V65%-45% and FEV1 is shown in the top row, while the bottom row depicts the relation with DLCO. The left column (a) presents the data for a patient with a ”positive” correlation: the decline of FEV1 and DLCO three months after the end of RT (t3m) is paralleled by an increase in CT density, which almost fully recovers three months later. The right column (b) shows a patient with a ”negative” relation: the improvement of FEV1 and DLCO three months after the end of RT is accompanied by an increase in CT density.
- Fig. 3 is of particular value to our understanding of post irradiation lung changes. We thank the reviewer for this remark.
- Pg 9 line 263, the fault could well be mine, but I didn't understand "As FEV1 was introduced into the radio-oncological literature based on surgery data this finding may reflect the fact that 91 patients in this cohort were operated." We modified the sentence as follows:
As 91 patients in this cohort were operated, the higher CC between FEV1 and CT morphology changes can be explained by the fact that a decrease in FEV1 represents a reduction primarily of the airways, which is the case when parts of the lung including bronchi and bronchioli are surgically removed.
- Line 273 is a strong point. This matter would best be in its own paragraph. The authors are arguing a clinical practice matter, and correctly in my opinion, showing that PFT is required prior to RT. Authors, please make this point clearer and in its own separate paragraph. Also restate your conclusion in the Abstract. We clarified this issue in a separate paragraph. Also, we re-stated the conclusion in the abstract.
Line 273. The current analysis clearly demonstrated a highly significant correlation of DLCO dynamics with the size of a specific radiation volume. Of note, this correlation remained statistically significant after Bonferroni correction for multiple testing (Figure 5), which allows to strongly argue in favor of pre-therapeutic PFTs as a pre-requisite for safe thoracic RT. Although this issue is a matter of ongoing debate, our results are in line with other studies in the field (reviewed by Niezink et al.).
Abstract, conclusion: The current analysis revealed a strong relation between the dynamics of DLCO and CT morphology changes in the mid-dose range, which is a strong indicator for the importance of routinely used PFTs in the context of a curative treatment approach.

Round 2
Reviewer 1 Report
The authors Adresse all questions an requests extraordinarily and at length.
In my opinion, the manuscript should be bublished.
Author Response
The authors Adresse all questions an requests extraordinarily and at length.
In my opinion, the manuscript should be bublished.
Again, we would like to thank reviewer 1 for his/her comments, which were very much appreciated.

Reviewer 2 Report
This manuscript has been much improved. The authors have eliminated most of the inconsistancies and confused prose. However minor grammar errors, run-on paragraphs, and linguistic oddities remain. These infelicities plus the many abbreviations in the Abstract again show a disregard for readers, an inattention to the importance of clear communication.
Although it has not been professionally edited, the manuscript is now acceptable. The core data the authors present is of value.
Author Response
This manuscript has been much improved. The authors have eliminated most of the inconsistancies and confused prose. However minor grammar errors, run-on paragraphs, and linguistic oddities remain. These infelicities plus the many abbreviations in the Abstract again show a disregard for readers, an inattention to the importance of clear communication.
Although it has not been professionally edited, the manuscript is now acceptable. The core data the authors present is of value.
Submission Date
23 February 2022
Date of this review
08 Apr 2022 18:14:34
Again, many thanks to reviewer 2 for going through the manuscript again. To the best of our knowledge we tried to adress the remaining issues in the revised manuscript.
run-on paragraphs
In order to improve the structure of the manuscript, we introduced paragraphs in the following lines: 318, 352, 369, 398.
Abstract
Line 5: measured in Hounsfield units (HU) was deleted.
Lines 14-15: with the automatically constrained deformation algorithm (ANACONDA) was deleted.
Lines 17: strong was replaced by significant
Line 19: The sentence was changed completely and runs as follows now: This volume range also predicted DLCO after RT (p-value 0.03).
Line 24: which is a strong indicator for was changed to convincingly indicates.
Introduction
Line 30: radiation instead of radiotherapy
Line 32: The beginning of the sentence was changed. The reason for this is the fact that the …
Line 41: considered was changed to regarded
Line 44: The words apparent and in density were deleted.
